# Exploring the Antidiabetic Properties of *Polyalthia longifolia* Leaf and Stem Extracts: In Vitro α-Glucosidase and Glycation Inhibition

**DOI:** 10.3390/molecules30214264

**Published:** 2025-10-31

**Authors:** Guglielmina Froldi, Marguerite Kamdem Simo, Laura Tomasi, Giulia Tadiotto, Francine Medjiofack Djeujo, Xavier Gabriel Fopokam, Emmanuel Souana, Modeste Lambert Sameza, Pierre Michel Jazet, Fabrice Fekam Boyom

**Affiliations:** 1Department of Pharmaceutical and Pharmacological Sciences, University of Padova, 35131 Padova, Italy; laura.tomasi@virgilio.com (L.T.); giuliatadiotto@gmail.com (G.T.); francine.medjiofackdjeujo@phd.unipd.it (F.M.D.); 2Antimicrobial & Biocontrol Agents Unit, Department of Biochemistry, University of Yaoundé I, Yaoundé P.O. Box 812, Cameroon; simomagui@yahoo.fr (M.K.S.); fopokam@gmail.com (X.G.F.); souana.emmanuel@facsciences-uy1.cm (E.S.); fabrice.boyom@fulbrightmail.org (F.F.B.); 3Department of Biological Sciences, University of Maroua, Maroua P.O. Box 814, Cameroon; 4Department of Biochemistry, University of Douala, Douala P.O. Box 24 157, Cameroon; samezamste@yahoo.com (M.L.S.); mjazet@yahoo.com (P.M.J.); 5Advanced Research & Health Innovation Hub, University of Yaoundé I, Yaoundé P.O. Box 337, Cameroon

**Keywords:** α-glucosidase inhibition, polyphenols, flavonoids, AGEs, traditional medicine, Annonaceae, ROS scavenging, HPLC-DAD analysis, antidiabetic activity

## Abstract

*Polyalthia longifolia*, a member of the Annonaceae family, is traditionally used for its medicinal properties, including as an antidiabetic remedy, primarily in Asia and sub-Saharan Africa. This study investigated the potential of six *P. longifolia* extracts in counteracting hyperglycemia and diabetes-related complications. Aqueous, ethanol, and methanol extracts from leaves and stems were evaluated for their antihyperglycemic, antiglycation, and antiradical properties using α-glucosidase, BSA, and ORAC assays, respectively. Phytochemical characterization was conducted using TPC and TFC assays, and HPLC analysis identified specific bioactive compounds, including various phenolic compounds (gallic acid, (+)-catechin, epicatechin, caffeic acid, ellagic acid and rosmarinic acid) and flavonoids (luteolin, kaempferol and baicalein). The MTT assay on the human cell line HT-29 assessed the activity of extracts on cell viability, showing slight cytotoxicity. Results demonstrated significant antidiabetic activity of the ethanol and methanol extracts from *P. longifolia* leaves. This study provides new insights into the potential use of *P. longifolia* in diabetes mellitus and supports the valorization of traditional medicinal plants.

## 1. Introduction

Diabetes mellitus (DM), a chronic metabolic disorder characterized by persistently high blood glucose levels, is a growing health concern. Projections indicate that by 2045, the global prevalence of DM will reach 783.2 million cases, a significant increase from the 537 million in 2021 [1]. Regrettably, this substantial rise is not expected to correspond with a proportional growth in healthcare expenditure. The majority of new cases are expected in developing countries with medium to low incomes, where DM-related healthcare costs are typically moderate [1]. For this reason, it is crucial to expand research on and improve knowledge about antidiabetic medicinal plants. These resources can provide more cost-effective treatment options, requiring less processing and infrastructure compared to synthetic or, moreover, biotechnological drugs, making plant-derived products particularly valuable in resource-limited conditions [2].

The genus *Polyalthia* includes 120 species, primarily found in Africa, Asia, and Australia. Native to southern India and Sri Lanka, *P. longifolia* has been subsequently introduced to various parts of Asia, and in West and Central Africa [3]. The species *P. longifolia* var. *pendula* (Sonn.) Thwaites, a member of the Annonaceae family, is a columnar tree that can reach a height of 20 m, characterized by lanceolate leaves with curvy margins and an acuminate apex, measuring between 12 and 30 cm in length (Appendix A). *P. longifolia* holds a significant place in Indian traditional medicine, where it is commonly employed for several medicinal purposes including treatment of hypertension, management of intestinal parasites, regulation of glycemia, and as anti-inflammatory remedy [4,5,6]. Phytochemical investigations reported the presence of various constituents, mainly terpenes (clerodane diterpenes), tannins, polyphenols, and aporphine alkaloids [5,6,7,8,9,10].

The literature reports few investigations regarding the antidiabetic activities of *P. longifolia*. Firstly, Nair et al. (2007) reported the capacity of *P. longifolia* leaf extracts to counteract hyperglycemia in alloxan-induced diabetic rats [11]. Successively, Sivashanmugam and Chatterjee (2013) described that ethanol and chloroform leaf extracts progressively reduced fasting blood glucose levels during a 28-week treatment, with effects comparable to metformin [12]. Furthermore, the authors reported that the leaf extracts were able to moderately reduce the activity of α-glucosidase and α-amylase enzymes in vitro [12,13]. However, the available data are limited and still require further investigation.

Hyperglycemia-induced damage is associated with various mechanisms; among these are insulin resistance, formation of radical species, and Advanced Glycation End Products (AGEs) [14]. These reactive compounds originate from non-enzymatic reactions between reducing sugars and macromolecules such as proteins, nucleic acids, or lipids. The process begins with the Maillard reaction, where an aldehyde or ketone carbonyl group of a reducing sugar (e.g., glucose, fructose, ribose) reacts with lysine and arginine amino acid residues [15]. Through various rearrangements and consequent reactions, this leads to the formation of AGEs, a heterogeneous class of irreversible compounds responsible for permanent cellular and tissue damage [16].

To evaluate the antidiabetic potential of *P. longifolia*, based on previously reported studies [11,12,13], this research examined six different extracts derived from either leaves (L) or stems (S) of *P. longifolia* (PL), prepared using aqueous (W), ethanol (E), and methanol (M) as solvents. The resulting abbreviations are: PLLW, PLLE, and PLLM for leaf extracts, and PLSW, PLSE, and PLSM for stem extracts. Their hypoglycemic potential was evaluated with the α-glucosidase assay, while antiglycation activity was estimated using the ribose-induced BSA (Bovine Serum Albumin) assay. Antioxidant capacity was determined by the ORAC (Oxygen Radical Absorbance Capacity) assay. Phytochemical composition was characterized through the detection of TPC (Total Phenol Content) and TFC (Total Flavonoid Content), as well as HPLC-DAD analysis. Furthermore, the safety of the extracts was explored through the MTT assay, detecting the cell viability on HT-29 cells. This approach aimed to provide an in-depth understanding of *P. longifolia* extracts for their potential medicinal use.

## 2. Results

### 2.1. In Vitro Antidiabetic Properties

#### 2.1.1. α-Glucosidase Inhibition

The yeast α-glucosidase assay was employed to evaluate the potential of *P. longifolia* extracts to inhibit the activity of the enzyme responsible for food carbohydrate digestion, which contributes to hyperglycemia in vivo. This assay serves as a model for the action of human α-glucosidase [17]. The reaction was monitored for 70 min under controlled conditions. Acarbose was used as positive control. Aqueous, ethanol, and methanol leaf and stem extracts were tested at concentrations ranging from 1 µg/mL to 100 µg/mL (Figure 1). Particularly, ethanol and methanol leaf extracts exhibited statistically significant inhibition at concentrations as low as 5 μg/mL (*p* < 0.01 and 0.05, respectively). In contrast, the aqueous leaf extract displayed a moderate inhibitory activity, only at the higher concentration of 25 μg/mL (19.55 ± 4.65%, *p* < 0.05). At this same concentration, both ethanolic and methanolic leaf extracts achieved near-complete enzyme inhibition (Figure 1). All stem extracts showed lower potency. The aqueous stem extract did not exhibit any inhibitory activity across all tested concentrations. The ethanol stem extract started showing significant inhibition at 25 μg/mL (15.63 ± 2.66%, *p* < 0.05), while the methanol stem extract only demonstrated inhibitory effects at 50 μg/mL and above. The maximum inhibition observed for 100 μg/mL *P. longifolia* ethanol and methanol stem extracts was 35.30 ± 2.61% and 51.25 ± 4.49%, respectively.

These findings suggest a marked difference in the α-glucosidase inhibitory effect between leaf and stem extracts of *P. longifolia*, with leaf extracts, particularly those obtained using ethanol and methanol as solvents, showing the most promising inhibition of enzyme activity.

The differential potency of *P. longifolia* extracts in inhibiting α-glucosidase activity is further proved using heatmap analysis (Figure 2). This visualization illustrates the dynamic changes in enzyme activity over a 70-min period, at various concentrations of each extract. The heatmap provides a color-coded representation of enzyme activity, where darker (red) shades indicate higher levels of enzyme activity, while lighter (clear blue) shades represent no enzyme action or inhibition over time. The relative white color represents an intermediate level of activity between clear blue and red. This analysis corroborates the findings previously reported in Figure 1, emphasizing the superior inhibitory potency of leaf extracts compared to stem extracts during the 70-min recording period. It clearly delineates the temporal patterns of inhibition, highlighting which extracts maintain their inhibitory effects over time and at what concentrations (Figure 2).

Based on the analysis of α-glucosidase inhibition and the respective IC_50_ values of each *P. longifolia* extract (Table 1), a clear potency hierarchy emerges: PLLM > PLLE > PLLW > PLSM > PLSE >> PLSW. This order demonstrates that leaf extracts consistently outperform stem extracts, with methanol-based leaf extract (PLLM) exhibiting the highest inhibitory potency. The substantial difference in efficacy between leaf and stem extracts underlines the importance of plant part selection in phytochemical studies. Consequently, for future investigations into α-glucosidase inhibition, these data strongly support prioritizing leaf extracts of *P. longifolia*, with a particular focus on methanol and ethanol-based preparations due to their higher inhibitory effects.

To further characterize the kinetics of α-glucosidase inhibition obtained with *P. longifolia* extracts, a graph of the slope of inhibition against logarithmic concentration of the extracts was obtained (Figure 3). This analysis allows for the determination of the Hill coefficient (nH), a parameter that describes the cooperativity in binding between the enzyme and inhibitor, and provides insights on the mechanism of enzyme inhibition [18].

The nH values for the leaf extracts PLLW, PLLE, and PLLM were found to be 2.44, 3.02, and 2.65, respectively (Table 1). These values, being greater than 1, suggest positive cooperativity among the components of each extract in the inhibition of α-glucosidase. This indicates that the binding of inhibitor molecules to the enzyme may enhance the binding of subsequent inhibitor molecules. The steepest Hill slope was observed for PLLE (nH = 3.02), suggesting that the ethanol leaf extract exhibits the highest degree of positive cooperativity among the extracts. This could imply that the constituents in PLLE may interact with multiple sites on the enzyme or induce conformational changes that enhance subsequent inhibitor binding. In contrast, the stem extracts showed minimal to no inhibition, likely due to lower concentrations of active compounds, precluding meaningful Hill plot analysis. The positive cooperativity observed in the leaf extracts, particularly in PLLE and PLLM, may contribute to their potent inhibitory effects on α-glucosidase, as reflected in their low IC_50_ values.

#### 2.1.2. Albumin Glycation Inhibition

Chronic hyperglycemia and concomitant oxidative stress promote non-enzymatic glycation reactions, leading to the formation of AGEs [19]. These can compromise the functionality of proteins, with albumin being particularly susceptible. In fact, the glycation of albumin is implicated in the pathogenesis of various vascular complications associated with DM, contributing to endothelial dysfunction [20].

To evaluate the antiglycation activity of leaf and stem extracts, the BSA assay was performed. This assay utilizes bovine serum albumin, which shares 76% sequence homology with the human protein, making it a suitable model for studying protein glycation [21]. This assay quantifies the capacity of any substance to inhibit the formation of AGEs between albumin and a reducing sugar, such as ribose [21]. The protocol involved an incubation period of 14 days of 10 mg/mL BSA with 0.05 M ribose. During this period, glycation values increase over time until stabilizing after 12–14 days (Appendix A). Aminoguanidine was used as positive control.

*P. longifolia* extracts were used at concentrations of 25, 50, 100, 250, and 500 µg/mL, since lower concentrations were not active. It was observed that the aqueous leaf extract (PLLW) does not exhibit any significant activity, whereas both ethanol (PLLE) and methanol (PLLM) extracts, at concentrations of 250 and 500 µg/mL, significantly inhibit BSA glycation (Figure 4). After 7 days of incubation, 500 µg/mL PLLE and PLLM inhibited ribose-induced glycation by 34.05 ± 0.66% and 43.56 ± 2.97%, respectively, showing a significant reduction in AGE formation. The ethanol (PLSE) and methanol (PLSM) stem extracts exhibited significant activity even at a concentration of 100 µg/mL, with inhibition rates of approximately 12% and 16%, respectively. After 7 days of incubation, 500 µg/mL PLSE and PLSM inhibited ribose-induced glycation by 43.34 ± 2.03% and 41.83 ± 3.65%, respectively. Under the experimental conditions, it was observed that the extracts maintain their moderate inhibitory action against albumin glycation throughout the entire 14-day incubation period.

### 2.2. Total Phenol and Flavonoid Contents

Plant-derived phenolic compounds, which include flavonoid compounds, are crucial components of curative plants due to their several health benefits, including antidiabetic effects [22,23]. This study evaluated the presence of these compounds in *P. longifolia* extracts, characterizing each type of extract using TPC and TFC assays [24]. Figure 5 shows the levels of TPC and TFC, comparing their amount in extracts obtained by use of water, ethanol, and methanol as solvent of extraction (Figure 5A,C), and between leaves and stems (Figure 5B,D). The methanol leaf extract (PLLM) exhibited the highest phenolic content among all the extracts; see Table 2. The relative phenolic content followed the order: PLLM > PLSM ≥ PLLE ≥ PLSE >> PLSW ≥ PLLW. These results demonstrate that methanol is the most appropriate solvent for the extraction of the phenolic compounds of *P. longifolia* leaves and stems, while water is the least effective, irrespective of the type of tissue (Figure 5). Regarding TFC, flavonoids are present in relatively low quantities, with higher content observed in PLLE and PLLM. Indeed, the TFC/TPC ratios clearly show that in these two extracts, the percentage of flavonoids is more relevant compared to the stem extracts and aqueous leaf extract (Table 2). Moreover, the varying TFC/TPC ratios across different extracts indicate that targeted extraction methods could be developed to obtain extracts enriched in specific phenolic subclasses.

### 2.3. Antiradical Scavenging Activity

It is well known that ROS formation is implicated in the progression of DM complications by several molecular mechanisms, even increasing the formation of AGEs [25,26]. Thus, the use of antioxidants can be considered a preventive intervention to reduce hyperglycemic-related diseases [25,27]. Considering the aim to find support for the use of *P. longifolia* against DM complications, the ORAC assay was conducted. This is a fluorometric assay that quantifies the antioxidant capacity of a substance based on its ability to neutralize peroxyl radicals by donating a hydrogen atom [28]. The results revealed higher antioxidant activity of both ethanol and methanol extracts derived from leaves and stems compared to aqueous extracts (Figure 6). The ethanol stem extract (PLSE) showed a TEAC value of 2130 ± 235 μmol TE/g extract, similar to that of the methanol stem extract, both significantly higher than all others (Table 2). Particularly, the stem extracts exhibited TEAC/TPC ratios more than twofold higher than those of leaf extracts, indicating that the phenolic compounds obtained from stems are more efficient as antiradical agents.

### 2.4. HT-29 Viability

The HT-29 cell line, derived from human colorectal adenocarcinoma, exhibits characteristics typical of mature intestinal cells, making it a valuable model for studying the potential cytotoxicity of plant extracts on human intestinal epithelium [29,30]. Based on this, the effects of leaf and stem *P. longifolia* extracts were tested at concentrations ranging from 1 to 250 µg/mL on HT-29 cell proliferation (Figure 7). The positive control of 10 µg/mL luteolin consistently reduced cell proliferation to 32.54 ± 1.14%.

The results showed that leaf and stem extracts demonstrated similar effects, while significant differences were observed when comparing aqueous extracts to ethanol or methanol extracts. The latter produced antiproliferative activity at concentrations ≥ 25 µg/mL, causing a concentration-dependent inhibition of cell viability (Figure 7). In contrast, the aqueous extracts (PLLW and PLSW) showed no detectable effect on cell proliferation up to 250 µg/mL (Figure 7). The potency order based on the IC_50_ values was: PLLM ≥ PLLE > PLSE > PLSM >> PLLW ≥ PLSW (Table 3). The nH provided insights into the concentration-effect relationships: PLLE, PLSE and PLLM have absolute values slightly higher than 1 (from 1.11 to 1.36), indicating a response proximal to a standard interaction between a compound and the target in the cells: one compound molecule, one effect on the cell. PLSM exhibited a more gradual curve (|nH| = 0.76), suggesting a potentially more complex interaction with the cells (Table 3).

These findings suggest a moderate antiproliferative activity of methanol and ethanol *P. longifolia* extracts on HT-29 cells, with leaf extracts showing higher potency than stem extracts. Aqueous extracts, however, did not influence HT-29 cell proliferation.

### 2.5. Phytochemical Characterization

The composition of *P. longifolia* leaf and stem extracts was analyzed using HPLC-DAD analysis. Chromatographic profiles revealed that the ethanol and methanol extracts of both leaves and stems exhibited a greater number of peaks with higher intensity compared to their aqueous counterparts (Figure 8). The analysis employed 11 polyphenolic standards for identification: baicalein, caffeic acid, (+)-catechin, chlorogenic acid, ellagic acid, epicatechin, gallic acid, kaempferol, luteolin, quercetin, and rosmarinic acid. These standards permitted the revelation of a diverse phytochemical profile across different solvents and plant parts (Table 4). Among phenolic compounds, gallic acid was exclusively detected in leaf methanol extract (PLLM), while (+)-catechin and epicatechin were found in all alcoholic extracts of both leaves and stems. Caffeic acid was identified in leaf water (PLLW), leaf ethanol (PLLE), and stem water (PLSW) extracts. Ellagic acid was detected across all extracts except the stem water extract (PLSW), with particularly high concentrations observed in PLLE and PLLM. Rosmarinic acid was prominently found in both leaf and stem ethanol extracts (PLLE and PLSE). Among flavonoids, luteolin was detected in low amounts in ethanol and methanol leaf extracts, kaempferol in ethanol and methanol stem extracts, while baicalein was consistently present across all extracts, regardless of solvent or plant part. These findings underscore the critical influence of the extraction solvent and plant tissue selection on the phytochemical extract profiles. The phenolic compounds detected in this HPLC-based phytochemical characterization are in agreement with data reported in previous studies [31,32,33,34,35]. These compounds themselves or together with other constituents could contribute to the observed in vitro antidiabetic activities reported. In particular, catechins, epicatechin, ellagic acid and luteolin, mainly identified in PLLE and PLLM in the antihyperglycemic and antiradical activities observed in this investigation [36,37,38,39,40,41,42]. Undoubtedly, further investigation is necessary for a deeper phytochemical characterization of *P. longifolia* extracts, mainly PLLM and PLLE, for their higher α-glucosidase inhibition, also searching for other types of constituents such as terpenes and alkaloids, which have also been reported in *P. longifolia* leaf extracts and could contribute to their activity [6,43].

## 3. Discussion

Various botanical products are widely proposed for therapy in medicine and as food supplements in many countries for DM management. However, only a fraction of these herbal remedies has undergone rigorous scientific study. This gap underscores the critical need for targeted research in this field. In fact, in vitro and in vivo studies have identified numerous plant-derived products with potential hypoglycemic activity. For example, *Momordica charantia* L. (bitter melon) and *Gymnema sylvestre* (Retz.) Schult. have shown promising results [44]. Particularly, *Galega officinalis* L. led to the discovery of metformin, a cornerstone of antidiabetic treatment [45]. The exploration of traditional medicinal plants, therefore, represents a valuable approach in the search for DM therapy.

In this study, yeast α-glucosidase served as a model assay for screening the potential of *P. longifolia* extracts as inhibitors of the enzyme activity, potentially helping to reduce glucose absorption from dietary carbohydrates. The leaf extracts exhibited significant concentration-dependent inhibition across all three solvents used for extraction. Specifically, the aqueous extract (PLLW) demonstrated significant inhibitory effects from the concentration of 25 µg/mL, while both ethanol (PLLE) and methanol (PLLM) extracts inhibited at concentrations as low as 5 µg/mL, indicating their higher potency. The respective EC_50_ values were: 36.87, 10.44, and 7.31 µg/mL (Table 1). Conversely, the stem extracts displayed substantially reduced potency. The aqueous stem extract (PLSW) showed no discernible inhibitory activity, while the ethanol (PLSE) and methanol (PLSM) stem extracts elicited only moderate inhibition at 50 µg/mL and above. Thus, PLLM extract emerged as the most potent inhibitor of α-glucosidase (IC_50_ = 7.31 µg/mL), exhibiting a positive cooperative interaction with the enzyme, as evidenced by a Hill slope greater than 1, suggesting potential allosteric modulation. Similar effects were shown by the ethanol and aqueous leaf extracts, but with lower potency.

Very few investigations have examined the inhibitory effects of *P. longifolia* preparations on α-glucosidase and α-amylase activities [12]. These studies utilized ethanol and chloroform [12] or methanol and petroleum ether leaf extracts [13], showing that alcohol extracts had superior inhibitory potency, in agreement with current results. Moreover, these extracts exhibited concentration-dependent inhibition of α-glucosidase at concentrations similar to those employed in the present investigation, but with lower potency [12,13]. Thus, present results not only support the potential of *P. longifolia* as a source of α-glucosidase inhibitors but also allow for the comparison of the efficacy of different solvents and tissues used for the extractions. Methanol and ethanol extracts of *P. longifolia* leaves promote significant concentration-dependent inhibition of α-glucosidase activity. Moreover, the leaf aqueous extract, important also for its traditional use, showed detectable inhibition. To date, no investigations related to the effects of aqueous extracts of *P. longifolia* on α-glucosidase activity have been found in the previous literature. Indeed, also few previous in vivo studies have reported the antihyperglycemic effects of *P. longifolia* extracts [11,12,46]. In one study, hyperglycemic rats with blood glucose levels ranging from 250 to 550 mg/dL showed an antihyperglycemic response after a 7-day treatment with methanol leaf extracts (300 mg/kg) [11]. Additionally, in a study using streptozotocin-induced diabetic rats, administration of ethanol and chloroform extracts (100 and 200 mg/kg) caused a significant reduction in fasting blood glucose levels, with effects observed from day 7 to day 28 of treatment [12]. In general, the literature data show that ethanol and methanol extracts from leaves of various medicinal plants, including Annonaceae species, are mostly active against α-glucosidase activity, suggesting their potential in diabetes mellitus treatment [47,48,49]. Our findings with *P. longifolia* align with this trend, as we observed significant α-glucosidase inhibition with both methanol and ethanol leaf extracts. Persistent hyperglycemia and oxidative stress promote non-enzymatic glycation reactions, leading to the formation of AGEs. These processes significantly alter the structures and functions of plasma proteins, particularly serum albumin, which plays a crucial role in the pathogenesis of diabetic vascular complications [19]. In the present study, a ribose-induced BSA glycation assay was employed to evaluate the antiglycation potential of *P. longifolia* extracts, tested in a concentration range of 25–500 µg/mL. Both leaf and stem extracts demonstrated moderate antiglycation activity, which was sustained throughout the 14-day incubation period. Among the leaf extracts, the aqueous preparation (PLLW) showed no inhibitory activity. In contrast, the ethanol (PLLE) and methanol (PLLM) leaf extracts exhibited comparable glycation inhibition at the higher concentrations, with PLLM achieving a maximum inhibition of 44% while PLLE reached 35%. Interestingly, the stem extracts displayed unexpectedly higher potency; both PLSE and PLSM extracts demonstrated significant antiglycation activity starting from 100 µg/mL, reaching a maximum inhibition of 44% at the higher concentrations. This level of inhibition was comparable to that observed with the leaf extracts, suggesting that *P. longifolia* stems may be an equally valuable source of antiglycation compounds. Previous studies have reported the antiglycation activity of various flavonoid fractions obtained from *P. longifolia* leaves using fructose-BSA and methylglyoxal-BSA assays, showing significant inhibition of AGE formation from 5 to 25 mg/mL [34].

Indeed, the TPC and TFC assays revealed a substantial presence of phenols and flavonoids in the *P. longifolia* extracts. The methanol leaf extract exhibited the highest TPC value (54.74 ± 1.90 mg GAE/g), while the flavonoid content was relatively low in all extracts. In PLLE and PLLM, the flavonoid content was 16.24% and 11.20% lower than the phenol content, respectively. A previous study reported the presence of phenols and flavonoids in methanolic extracts of *P. longifolia* leaves, with values of TPC = 87.43 ± 1.23 mg GAE/g and TFC = 70.25 ± 3.12 mg catechin equivalent/g [50]. The differences observed could be attributed to different extraction protocols and also to differences in geographical origin and growing conditions. Environmental factors such as soil composition, climate, and altitude can significantly influence the biosynthesis and accumulation of secondary metabolites in plants [51].

In the current investigation, a consistent correlation between phenolic content and extraction solvent was observed for both leaf and stem extracts. Specifically, aqueous extracts (PLLW and PLSW) exhibited lower phenolic content, while methanolic extracts (PLLM and PLSM) exhibited higher levels. This can be explained by the fact that methanol can effectively extract both moderately polar and some less polar compounds, whereas highly polar water will primarily extract highly polar compounds. In general, the higher amount of phenolic and flavonoid compounds in methanol and ethanol extracts suggests that these solvents may be more effective in extracting these bioactive compounds from *P. longifolia* tissues. This information is valuable for optimizing extraction procedures in future studies and potential therapeutic applications.

Moreover, the significant presence of phenolics and flavonoids in *P. longifolia* extracts correlates with their antioxidant properties, as quantified by ORAC assay. This analysis revealed consistently higher TEAC values for stem extracts compared to leaf extracts across all the extraction solvents employed. Particularly, the TEAC/TPC ratios were markedly higher for stem extracts (50–67) than for leaf extracts (26–32, Table 2), suggesting that the phenolic compounds in stem extracts have the highest antioxidant capacity. These findings expand previous research, which reported good antiradical activity of a methanol leaf extract using the DPPH assay [50].

In this study, the cytotoxicity of *P. longifolia* extracts (1 to 250 µg/mL) on HT-29 cell viability was investigated using the MTT assay. The aqueous extracts of both leaves and stems (PLLW and PLSW) showed no significant effects on cell growth. However, the ethanol and methanol extracts caused moderate cytotoxicity at concentrations ≥ 25 µg/mL. The potency order of cell proliferation inhibition based on IC_50_ values was: PLLM ≥ PLLE > PLSE > PLSM >> PLLW ≥ PLSW. The extracts PLLE, PLSE, and PLLM exhibited absolute nH values slightly higher than 1 (ranging from 1.11 to 1.36), indicating a response proximal to a standard interaction between a compound and its cellular target. Otherwise, PLSM demonstrated a more gradual curve (|nH| = 0.76), suggesting a potentially more complex interaction with the cells. Based on the current results, the aqueous extracts of *P. longifolia* appear to have a very favorable safety profile, showing no significant cytotoxicity. Differently, the ethanol and methanol extracts had moderate cytotoxicity, suggesting they may be used at lower doses. While these in vitro results provide initial insights, comprehensive toxicological studies, including in vivo experiments, are necessary to establish the safety of these extracts for potential phytotherapeutic applications.

Previous phytochemical studies on *P. longifolia* have identified various constituents, including alkaloids (particularly clerodane diterpenes), terpenoids, steroids (β-sitosterol and stigmasterol) [52,53,54,55], flavonoids [32,33,56], tannins, saponins, phenolic compounds, as gallic acid [33,57,58]. Flavonoids like quercetin, rutin, and chrysin are known for their antioxidant and anti-inflammatory properties, while the isoflavone daidzein exhibits estrogenic and antidiabetic effects [33,56,59]. In the current investigation, HPLC analysis revealed distinct chromatographic profiles for leaf and stem extracts. The aqueous extracts showed fewer constituents, while the ethanol and methanol extracts demonstrated a richer phytochemical composition. The analysis identified various phenolic compounds and flavonoids, with distribution varying by solvent and plant tissue. Gallic acid was exclusive to leaf methanol extract, while (+)-catechin and epicatechin were present in all alcoholic extracts. Caffeic and ellagic acids showed selective distribution, and baicalein was ubiquitous among extracts. The results emphasize the critical influence of extraction methods and plant tissue selection on phytochemical profiles, highlighting the importance of solvent choice in extracting bioactive compounds from *P. longifolia*. The presence of these phenolic compounds and flavonoids, particularly catechins, epicatechins, and baicalein, may contribute to the observed α-glucosidase inhibition, antiglycation effects, and antioxidant activity of *P. longifolia* extracts, as these compounds have been previously associated with such beneficial properties [30,60,61]. While the present study does not establish a direct causal relationship between these compounds and the observed antidiabetic effects, their presence in the most active extracts (PLLE and PLLM) suggests that they may play a significant role in the biological activities observed. Future studies involving the isolation and testing of these individual compounds could provide more definitive evidence of their contribution to the antidiabetic properties of *P. longifolia* extracts. Moreover, several other types of constituents may have a role in these activities.

The comparison between α-glucosidase inhibition and cytotoxicity data reveals that *P. longifolia* leaf extracts, particularly the methanol (PLLM) and ethanol (PLLE) extracts, show the most promise for potential applications. These extracts demonstrate high α-glucosidase inhibitory activity at concentrations below cytotoxic levels, indicating a favorable therapeutic window. PLLM exhibits the best balance of high inhibitory potency (IC_50_ = 7.31 µg/mL) and low cytotoxicity (observed at ≥ 25 µg/mL), making it the most promising candidate for further investigation. The aqueous leaf extract (PLLW) shows a good safety profile but lower potency, which might be suitable for applications requiring milder effects and in phytotherapy. In contrast, stem extracts appear less promising due to their lower inhibitory activity occurring at potentially cytotoxic concentrations. These findings suggest that future research should focus on leaf extracts, particularly PLLM and PLLE, to identify specific compounds responsible for the high α-glucosidase inhibitory activity. Additionally, extending these studies to in vivo models could provide valuable insights into the efficacy and safety of these extracts in more complex biological systems, potentially leading to the development of new plant-derived antidiabetic agents.

## 4. Materials and Methods

### 4.1. Chemical Reagents

Acarbose, aluminum chloride hexahydrate, aminoguanidine (AG), 2,2′-azobis(2-amidinopropane) dihydrochloride (AAPH), bovine serum albumin (BSA), fluorescein, Folin & Ciocalteu’s phenol reagent, gallic acid, 3-(4,5-dimethylthiazol-2-yl)-2,5-diphenyl-2H-tetrazolium bromide (MTT), fetal bovine serum (FBS), α-glucosidase (EC 3.2.1.20, *Saccharomyces cerevisiae* type I, 10 U/mg protein), 6-hydroxy-2,5,7,8-tetramethylchroman-2-carboxylic acid (Trolox), *p*-nitrophenyl-α-D-glucopyranoside (*p*-NPG), quercetin, ribose, sodium azide, and sodium carbonate were purchased from Merck Life Science (Milano, Italy). HPLC standards, all other chemicals and solvents were of analytical grade or higher purity (≥ 95% for reference standards, Merck Life Science). Purified water was obtained using a Milli-Q water purification system (Millipore, Burlington, MA, USA).

### 4.2. Plant Extract Preparations

Mature leaves and young woody stems (0.5–1 cm in diameter) of *P. longifolia* var. *pendula* were collected in June 2023 in Yaoundé (Centre Region, Cameroon). The plant material was identified at the National Herbarium of Cameroon by comparison with existing specimens and recorded under the code number 67475HNC.

The leaves and stems were dried at room temperature away from light, then ground into powder. For each extraction, 30 g of powder was used. Three solvents were employed: distilled water, ethanol, and methanol. Each powder sample was macerated in 0.5 L of solvent for 72 h at room temperature. This process was repeated three times to ensure a broad extraction of plant constituents. The resulting mixtures were filtered through Whatman No. 3 filter paper. Alcoholic extracts (methanol and ethanol) were rotary evaporated at 40 °C until complete solvent evaporation. The aqueous extracts were dried in an oven at 40 °C. Successively, the dried extracts were codified as PLLW, PLLE, PLLM (leaf extracts) and PLSW, PLSE, PLSM (stem extracts) for water, ethanol, and methanol, respectively. All extracts were stored at 4 °C.

The final yields were as follows: leaf extracts: 6.40% (water), 9.40% (ethanol), and 11.48% (methanol); stem extracts: 1.27% (water), 2.64% (ethanol), and 2.44% (methanol). Samples of the dried extracts from leaves (PLL1-3) and stems (PLS5-7) are deposited at the Department of Pharmaceutical and Pharmacological Sciences, Laboratory of Pharmacognosy, University of Padova (Italy).

### 4.3. HPLC-DAD Analysis

The chromatographic analysis was performed using a Waters system (Milan, Italy) equipped with a 1525 binary pump and a 2998 photodiode array detector. Separation was achieved on a Symmetry C18 column (4.6 × 75 mm, 3.5 µm, Waters). The mobile phase consisted of (A) 0.1% *v*/*v* acetic acid in water, and (B) 0.1% *v*/*v* acetic acid in acetonitrile, delivered at 1 mL/min. The gradient elution profile was as follows: 0–1 min: 95% A; 1–7 min: 95–75% A; 7–9 min: 75–60% A; 9–12 min: 60–55% A; 12–14 min: 55–50% A; 14–18 min: 50–40% A; 18–23 min: 40–20% A; 23–28 min: 20–95% A; 28–30 min: 95% A.

*P. longifolia* ethanol and methanol extracts were dissolved in methanol (2 mg/mL), while *P. longifolia* water extracts were solubilized in 50:50 water-methanol. The samples were filtered through a 0.22 µm filter before injecting a volume of 20 µL. Chromatograms were recorded from 210 to 400 nm, with specific analysis at 254 nm (phenolic compounds) and 340 nm (flavonoids). The compound identification was based on retention time and spectral matching with standards.

Eleven reference compounds were used for identification: baicalein, caffeic acid, (+)-catechin, chlorogenic acid, ellagic acid, epicatechin, gallic acid, kaempferol, luteolin, quercetin, and rosmarinic acid. Data acquisition and analysis were performed using Breeze 2 software.

### 4.4. Yeast α-Glucosidase Inhibition Assay

The α-glucosidase inhibitory activity was assessed using a previously reported method [41,42]. The assay was performed in triplicate in 96-well plates. Each well contained: 80 μL of sample extract solution or acarbose (positive control, used at the final concentration of 800 µg/mL), 20 μL of α-glucosidase solution (0.05 U/mL final concentration). The extracts were tested at final concentrations of 1, 5, 10, 25, 50 and 100 µg/mL. After 10 min of incubation at 37 °C, 100 μL of 4 mM *p*-NPG substrate was added to start the reaction (final concentration 2 mM). Enzyme activity was monitored at 405 nm for 70 min using a PerkinElmer Victor Nivo microplate reader (Waltham, MA, USA). Results were expressed as a percentage of the maximum enzyme activity observed without inhibitors.

### 4.5. BSA Glycation Inhibition Assay

The antiglycation activity of *P. longifolia* extracts was assessed using a ribose-induced BSA glycation assay [62,63]. BSA (10 mg/mL, pH 7.4) was incubated with 0.05 M ribose at 37 °C for 14 days in the dark, with or without the extracts at concentrations of 25, 50, 100, 250 and 500 µg/mL. Fluorescence intensity, indicative of AGE formation, was measured at excitation/emission wavelengths of 355/460 nm using a PerkinElmer Victor Nivo microplate reader (Waltham, MA, USA). Aminoguanidine (2.5 mM) served as positive control. Readings were taken on days 2, 5, 7, 9, 12, and 14 to monitor the glycation reaction over time. Results were expressed as a percentage of the maximum BSA fluorescence induced by ribose alone.

### 4.6. TPC and TFC Assays

The TPC assay, a colorimetric test for quantifying phenolic compounds in *P. longifolia* extracts, was performed using transparent 24-well plates [64]. Gallic acid solutions (25–500 µg/mL) and the extract solutions were prepared. Each well contained 1400 µL deionized water, 100 µL Folin–Ciocalteu’s reagent, and 20 µL of sample or gallic acid. After 8 min incubation, 300 µL saturated Na_2_CO_3_ solution was added to obtain an alkaline environment [65]. After 2 h of incubation at room temperature, absorbance was measured at 760 nm using a PerkinElmer Victor Nivo™ plate reader (Waltham, MA, USA). Total phenolic content was determined using a gallic acid calibration curve and expressed as mg gallic acid equivalents (GAE) per g of dry extract.

The TFC assay was performed in 24-well plates using a quercetin calibration series (6.25–100 µg/mL). Each well contained 250 µL of quercetin standard or the sample extract solution, 1130 µL methanol, and 120 µL AlCl_3_ (25% *w*/*v*). After 15 min incubation at room temperature, absorbance was measured at 410 nm using a PerkinElmer Victor Nivo™ plate reader (Waltham, MA, USA). Blank-subtracted absorbance values were plotted against quercetin concentrations to create a calibration curve. Total flavonoid content was determined by interpolation and expressed as mg quercetin equivalents (QE) per g of extract.

### 4.7. ORAC Assay

The ORAC assay was used to evaluate the ability of the extracts to counteract the peroxyl radical-induced oxidative reactions [66]. Fluorescein (0.08 µM) was mixed with the sample solutions, PBS (blank), and Trolox (6.25–50 µM, reference standard). Samples were incubated at 37 °C for 10 min. The oxidative reaction was initiated by adding 0.15 M AAPH. Fluorescence decay was monitored for 70 min at 37 °C using a PerkinElmer Victor Nivo microplate reader (Waltham, MA, USA). ORAC values were expressed as Trolox Equivalent Antioxidant Capacity (TEAC, µmol TE/g of extract).

### 4.8. HT-29 Cell Viability Assay

Cell viability was determined using the MTT assay [67]. Human Caucasian colon adenocarcinoma (HT-29) cells were cultured in RPMI 1640 medium supplemented with 10% FBS and maintained in an incubator at 37 °C (5% CO_2_) until reaching confluence in Petri dishes (Appendix A). Following trypsinization for detachment, the HT-29 cells were seeded in 96-well plates (5000 cells per well) and allowed to adhere overnight. The cultures were then treated with each *P. longifolia* extract at 1, 5, 10, 25, 50, 100, and 250 µg/mL, or with medium alone (control). After 24 h, the cells were washed with RPMI 1640 medium and subsequently treated with 0.05 mg/mL MTT for 4 h. During this period, viable cells reduce the tetrazolium reagent to produce purple formazan crystals, which were then solubilized with 2-propanol (Appendix A). Finally, absorbance was measured using a PerkinElmer Victor Nivo microplate reader (Waltham, MA, USA) at a wavelength of 570 nm. The mean absorbance (Abs) was calculated for each sample concentration and for the controls. The percentage of cell viability in the presence of various sample concentrations was calculated relative to the control.

### 4.9. Data Analysis and Statistical Methods

Data are presented as mean ± standard error of the mean (SEM) from 3 to 7 independent experiments. Sigmoid curve fitting and statistical evaluations were performed using GraphPad Prism 10.2.3 (San Diego, CA, USA). The half-maximum inhibitory concentration (IC_50_), minimum effective concentration (MEC, the lowest concentration showing significant inhibition, *p* < 0.05), and Hill slope coefficient (nH) were calculated by nonlinear regression [18,68,69]. The statistical significance between the control and each treatment was evaluated using Student’s *t*-test, while comparisons among three or more groups were performed with one-way ANOVA, followed by Tukey’s multiple comparison test. Statistical significance was set at *p* < 0.05.

## 5. Conclusions

This investigation explored the potential antidiabetic properties of *P. longifolia* extracts, focusing on leaf and stem preparations using various solvents. Leaf extracts, particularly methanol (PLLM) and ethanol (PLLE) preparations, exhibited significant α-glucosidase inhibition at non-cytotoxic concentrations, with PLLM showing the most promising balance of high inhibitory potency (IC_50_ = 7.31 µg/mL) and low cytotoxicity (IC_50_ = 24.12 µg/mL). The studies on HT-29 cell viability suggest that *P. longifolia* leaf extracts are potentially safe for oral administration, showing an antidiabetic window between α-glucosidase inhibition and cytotoxicity. Furthermore, both leaf and stem extracts displayed moderate antiglycation capacity—by inhibiting the formation of AGEs—and antioxidant capacity. The appreciable phenolic and flavonoid content in these extracts correlates with their antidiabetic and antioxidant properties. HPLC analysis identified various phenolic compounds and flavonoids, including gallic acid, catechins, epicatechins, caffeic acid, ellagic acid, rosmarinic acid, luteolin and baicalein, which may contribute to the observed biological activities.

The use of leaves, a renewable resource, coupled with the possibility of domestic cultivation, enhances the cost-effectiveness and sustainability of *P. longifolia* as a source of new antidiabetic medicines. Future studies should focus on isolating and characterizing specific bioactive compounds, particularly from leaf methanol and ethanol extracts. Furthermore, in vivo studies are necessary to evaluate the efficacy and safety, and clinical trials are crucial next steps to validate the potential therapeutic applications of *P. longifolia* in diabetes mellitus.

## Figures and Tables

**Figure 1 molecules-30-04264-f001:**
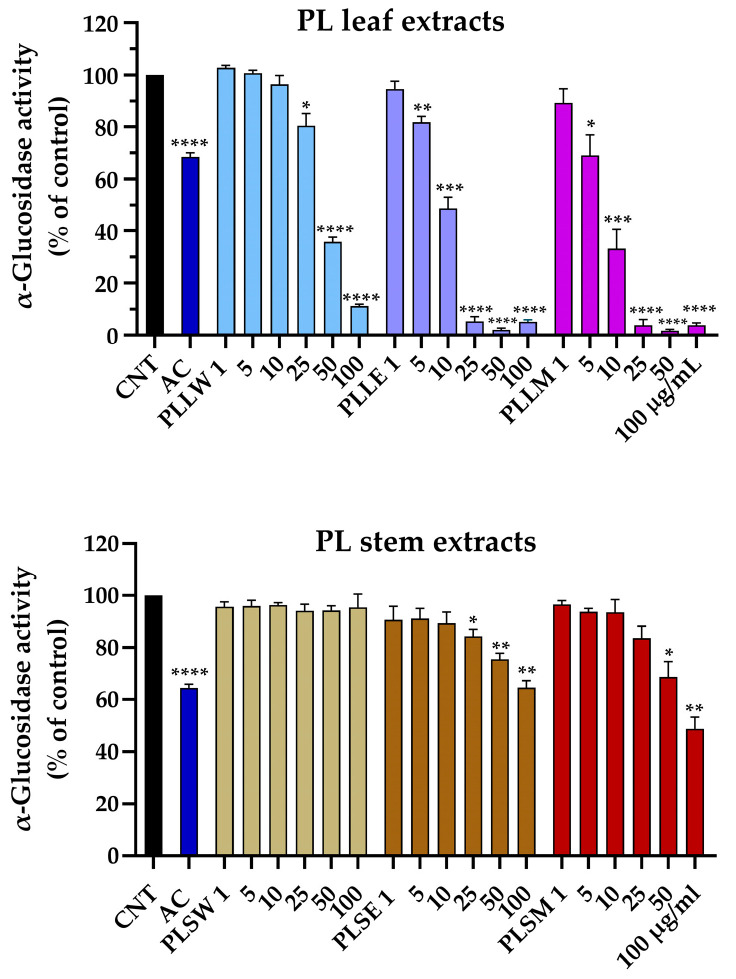
Effects of aqueous (W), ethanol (E), and methanol (M) extracts obtained from leaves and stems of *Polyalthia longifolia*, tested at concentrations ranging from 1 to 100 µg/mL, on α-glucosidase activity after 70 min of incubation. Leaf extracts are shown in shades of blue and violet (light blue, light violet, pink) and stem extracts in shades of maroon (light maroon, maroon, dark red) for W, E, and M extracts, respectively. AC: acarbose (800 µg/mL, positive control, blue column). Data are presented as mean ± SEM of 5 triplicate experiments. *: *p* < 0.05; **: *p* < 0.01; ***: *p* < 0.001; ****: *p* < 0.0001 vs. control (CNT). PLLW: *P. longifolia* leaf water extract; PLLE: *P. longifolia* leaf ethanol extract; PLLM: *P. longifolia* leaf methanol extract; PLSW: *P. longifolia* stem water extract; PLSE: *P. longifolia* stem ethanol extract; PLSM: *P. longifolia* stem methanol extract.

**Figure 2 molecules-30-04264-f002:**
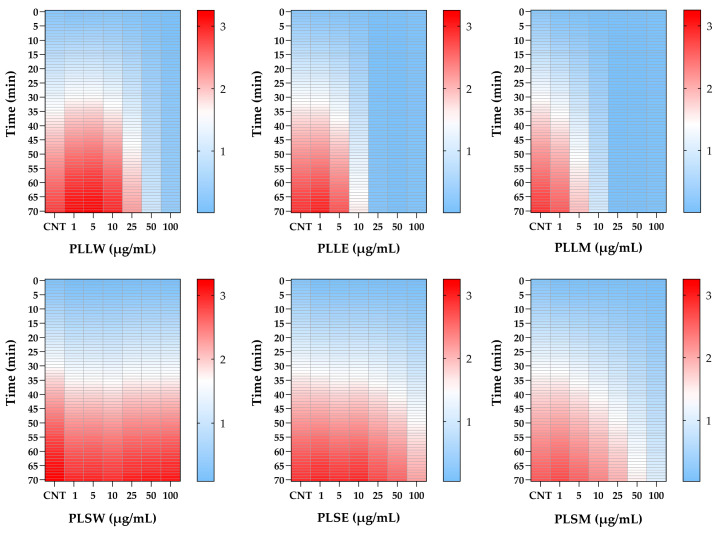
Heatmap analysis of α-glucosidase activity inhibited by *Polyalthia longifolia* extracts over a 70-min period, in the absence (CNT) and in the presence of leaf and stem extracts, at concentrations from 1 to 100 μg/mL. Color intensity represents enzyme activity levels: red indicates higher enzyme activity, white represents very low activity, while lighter blue shows no activity. The heatmap demonstrates the superior inhibitory potency of leaf extracts compared to stem extracts and allows for comparison of inhibitory effects across different extracts, concentrations, and time points. PLLW: *P. longifolia* leaf water extract; PLLE: *P. longifolia* leaf ethanol extract; PLLM: *P. longifolia* leaf methanol extract; PLSW: *P. longifolia* stem water extract; PLSE: *P. longifolia* stem ethanol extract. PLSM: *P. longifolia* stem methanol extract.

**Figure 3 molecules-30-04264-f003:**
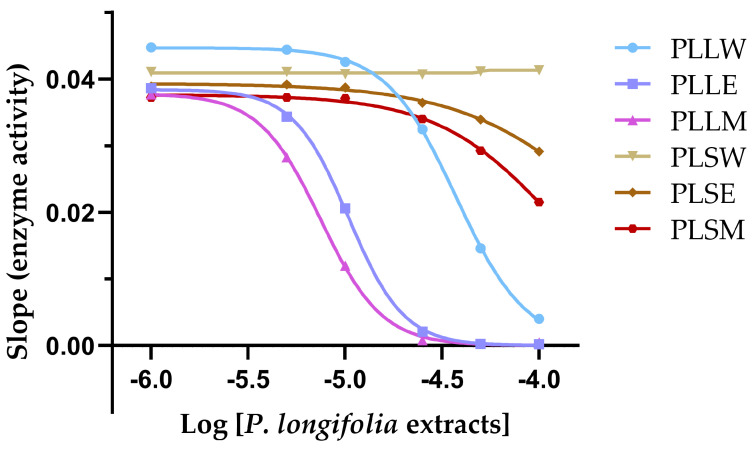
The slope of inhibition vs. the logarithmic concentrations of each *Polyalthia longifolia* extract (Hill plot) allows determination of the Hill coefficient (nH) and provides insights into the cooperativity of enzyme inhibition. PLLW: *P. longifolia* leaf water extract; PLLE: *P. longifolia* leaf ethanol extract; PLLM: *P. longifolia* leaf methanol extract; PLSW: *P. longifolia* stem water extract; PLSE: *P. longifolia* stem ethanol extract; PLSM: *P. longifolia* stem methanol extract.

**Figure 4 molecules-30-04264-f004:**
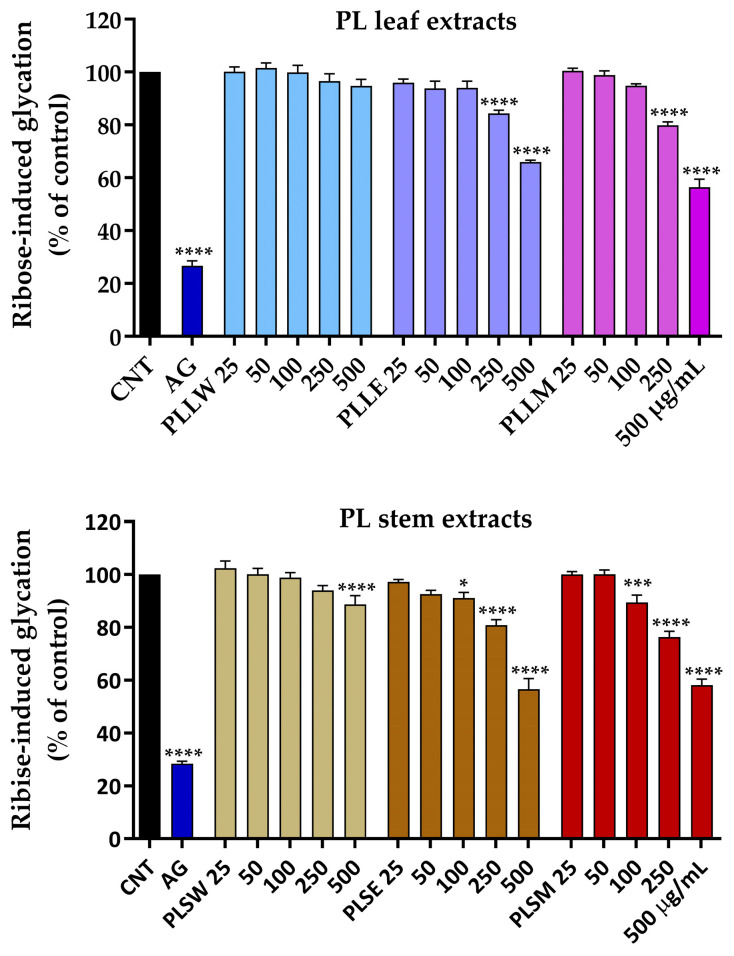
Effects of aqueous (W), ethanol (E), and methanol (M) extracts obtained from leaves and stems of *Polyalthia longifolia*, tested at concentrations ranging from 25 to 500 µg/mL, on ribose-induced BSA glycation after 7-day incubation at 37 °C. Leaf extracts are shown in shades of blue and violet (light blue, light violet, pink) and stem extracts in shades of maroon (light maroon, maroon, dark red) for W, E, and M extracts, respectively. Data are presented as mean ± SEM of 3 triplicate experiments. *: *p* < 0.05; ***: *p* < 0.001; ****: *p* < 0.0001 vs. control (CNT). AG: aminoguanidine (2.5 mM, positive control, blue column). PLLW: *P. longifolia* leaf water extract; PLLE: *P. longifolia* leaf ethanol extract; PLLM: *P. longifolia* leaf methanol extract; PLSW: *P. longifolia* stem water extract; PLSE: *P. longifolia* stem ethanol extract; PLSM: *P. longifolia* stem methanol extract.

**Figure 5 molecules-30-04264-f005:**
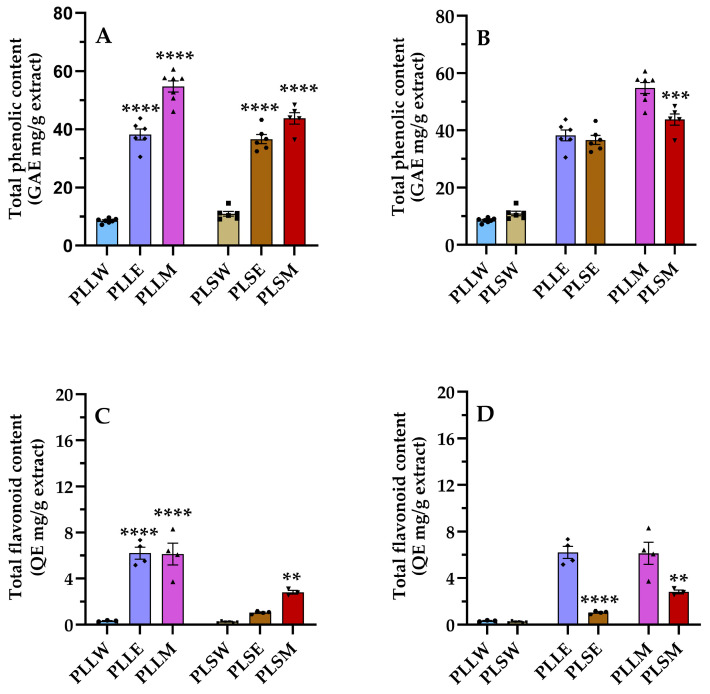
Phenolic (**A**,**B**) and flavonoid (**C**,**D**) contents of *Polyalthia longifolia* extracts. Panels A and C show the total phenolic content (TPC) and total flavonoid content (TFC) grouped by plant tissue type (leaf and stem). Panels (**B**,**D**) show TPC and TFC, grouped by solvent type (water, ethanol, methanol). Leaf extracts are shown in shades of blue and violet (light blue, light violet, pink) and stem extracts in shades of maroon (light maroon, maroon, dark red) for water (W), ethanol (E), and methanol (M) extracts, respectively. The symbols (circle, triangle, and square) represent the value of each experiment. Data are presented as mean ± SEM of 5–7 duplicate experiments. GAE: gallic acid equivalents expressed as mg/g extract; QE: quercetin equivalents expressed as mg/g extract. ****: *p* < 0.0001 vs. *P*. *longifolia* aqueous extracts (**A**,**C**); ***: *p* < 0.001 PLSM vs. PLLM (**B**); **: *p* < 0.01 PLSM vs. PLSW (**C**); ****: *p* < 0.0001 PLSE vs. PLLE (**D**); **: *p* < 0.01 PLSM vs. PLLM (**D**). PLLW: *P. longifolia* leaf water extract; PLLE: *P. longifolia* leaf ethanol extract; PLLM: *P. longifolia* leaf methanol extract; PLSW: *P. longifolia* stem water extract; PLSE: *P. longifolia* stem ethanol extract; PLSM: *P. longifolia* stem methanol extract.

**Figure 6 molecules-30-04264-f006:**
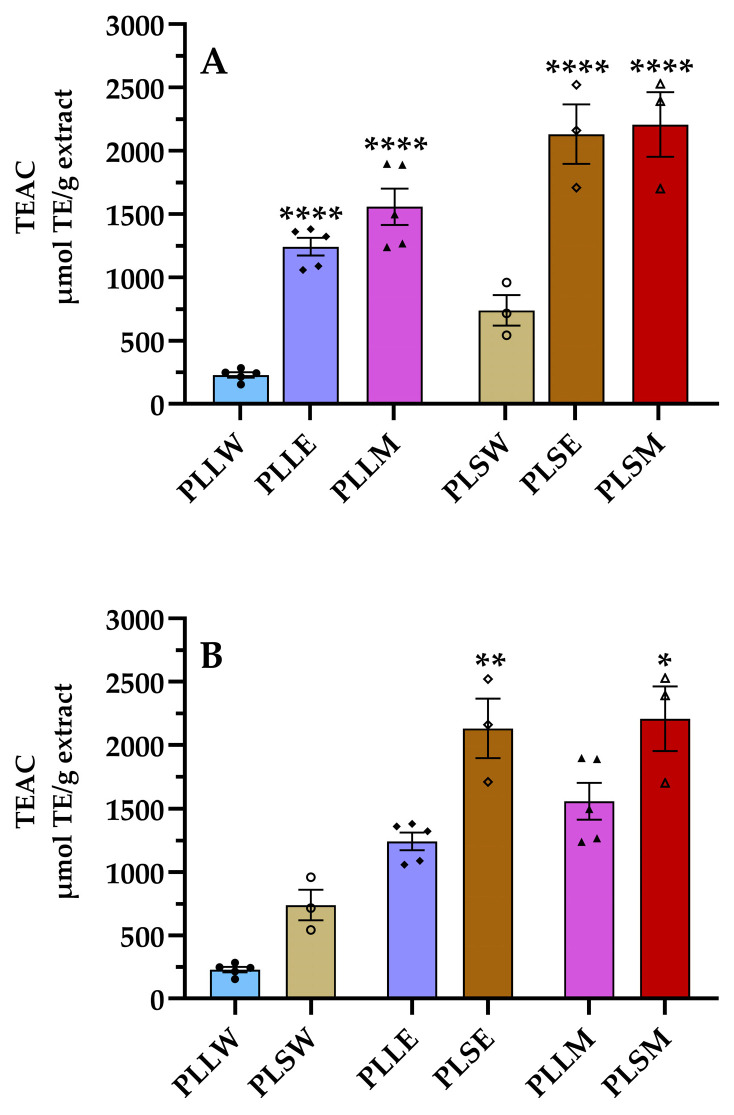
Antiradical activity of aqueous (W), ethanol (E), and methanol (M) extracts obtained from leaves and stems of *Polyalthia longifolia* determined using the oxygen radical absorbance capacity (ORAC) assay. Panel (**A**) shows a comparison based on tissue types, while Panel (**B**) compares extraction solvents. Leaf extracts are shown in shades of blue and violet (light blue, light violet, pink) and stem extracts in shades of maroon (light maroon, maroon, dark red) for W, E, and M extracts, respectively. The symbols (circle, triangle, and square) represent the value of each experiment. PLLW: *P. longifolia* leaf water extract; PLLE: *P. longifolia* leaf ethanol extract; PLLM: *P. longifolia* leaf methanol extract; PLSW: *P. longifolia* stem water extract; PLSE: *P. longifolia* stem ethanol extract; PLSM: *P. longifolia* stem methanol extract. TEAC: Trolox equivalent antioxidant capacity. Data are presented as mean ± SEM of 3–5 duplicate experiments. ****: *p* < 0.0001 vs. *P*. *longifolia* aqueous extracts; **: *p* < 0.01 vs. PLLE; *: *p* < 0.05 vs. PLLM.

**Figure 7 molecules-30-04264-f007:**
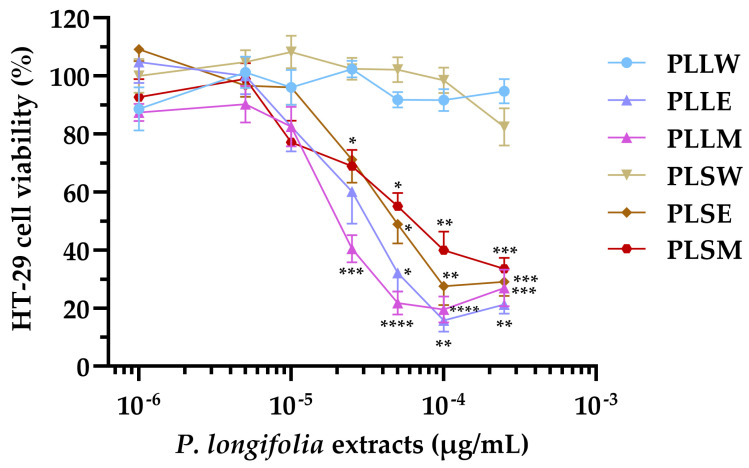
Effect of the *Polyalthia longifolia* extracts on HT-29 cell viability after 24 h of incubation. Data are presented as mean ± SEM of 5–6 triplicate experiments. ****: *p* < 0.0001; ***: *p* < 0.001; **: *p* < 0.01; *: *p* < 0.05 vs. control (untreated cells). PLLW: *P. longifolia* leaf water extract; PLLE: *P. longifolia* leaf ethanol extract; PLLM: *P. longifolia* leaf methanol extract; PLSW: *P. longifolia* stem water extract; PLSE: *P. longifolia* stem ethanol extract; PLSM: *P. longifolia* stem methanol extract.

**Figure 8 molecules-30-04264-f008:**
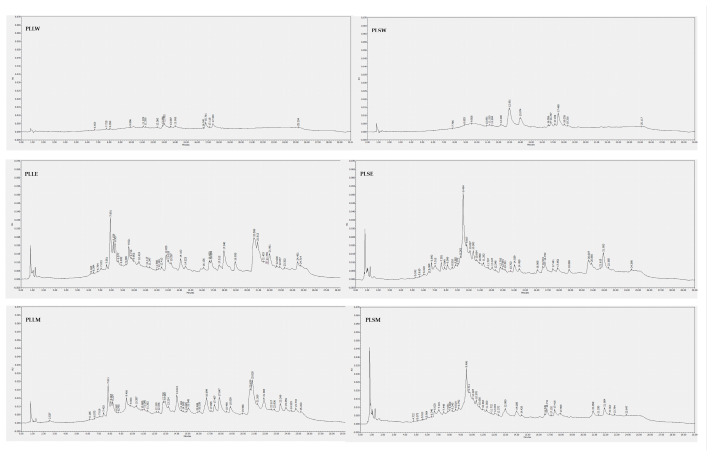
HPLC-DAD chromatograms of *Polyalthia longifolia* extracts. The chromatograms were obtained from the injection of 20 μg/mL of each extract dissolved in methanol, except for the water extracts of leaves and stems, which were solubilized in 50:50 water-methanol. Detection is reported at 254 nm. PLLW: *P. longifolia leaf* water extract; PLLE: *P. longifolia* leaf ethanol extract; PLLM: *P. longifolia* leaf methanol extract; PLSW: *P. longifolia* stem water extract; PLSE: *P. longifolia* stem ethanol extract; PLSM: *P. longifolia* stem methanol extract.

**Table 1 molecules-30-04264-t001:** α-Glucosidase inhibition obtained with *Polyalthia longifolia* leaf and stem extracts.

*P. longifolia* Extract	α-Glucosidase Inhibition IC_50_ µg/mL (95% CI)	α-Glucosidase Inhibition nH	Primary Effect
PLLW	36.87 (34.72–39.52)	2.44	Weak inhibition
PLLE	10.44 (9.66–11.32)	3.02	Potent inhibition
PLLM	7.31 (6.61–8.21)	2.65	Most potent inhibition
PLSW	--	--	No inhibition
PLSE	≥100	--	Very weak inhibition
PLSM	≥100	--	Very weak inhibition

PLLW: *P. longifolia* leaf water extract; PLLE: *P. longifolia* leaf ethanol extract; PLLM: *P. longifolia* leaf methanol extract; PLSW: *P. longifolia* stem water extract; PLSE: *P. longifolia* stem ethanol extract; PLSM: *P. longifolia* stem methanol extract. IC_50_: half maximal inhibitory concentration expressed in µg/mL; nH: Hill coefficient, calculated by non-linear regression analysis of concentration-enzyme inhibition (data of Figure 3).

**Table 2 molecules-30-04264-t002:** Total phenolic and flavonoid contents, and oxygen radical absorbance capacity detected in the leaf and stem *Polyalthia longifolia* extracts.

	PLLW	PLLE	PLLM	PLSW	PLSE	PLSM
TPC GAE mg/g	8.57 ± 0.30	38.24 ± 1.88 ****	54.74 ± 1.90 ****	10.98 ± 0.79	36.65 ± 1.57 ****	43.76 ± 2.01 ****
TFC QE mg/g	0.32 ± 0.03	6.21 ± 0.51 ****	6.13 ± 0.94 ****	0.26 ± 0.03	1.06 ± 0.06	2.81 ± 0.16 **
TFC/TPC (%)	3.73	16.24	11.20	2.37	2.89	6.42
TEAC µmol TE /g	228 ± 22	1242 ± 70 ***	1558 ± 144 ****	739 ± 121	2130 ± 235 ****	2207 ± 256 ****
TEAC/TPC µmol TE/mg GAE	26.60	32.48	28.46	67.30	58.12	50.43

PLLW: *P. longifolia* leaf water extract; PLLE: *P. longifolia* leaf ethanol extract; PLLM: *P. longifolia* leaf methanol extract; PLSW: *P. longifolia* stem water extract; PLSE: *P. longifolia* stem ethanol extract; PLSM: *P. longifolia* stem methanol extract. TFC/TPC: ratio of total flavonoid content to total phenolic content (expressed as a percentage); TEAC: Trolox equivalent antioxidant capacity; TEAC/TPC: ratio of antioxidant capacity to total phenolic content. Data are presented as mean ± SEM of 3–7 duplicate experiments. ****: *p* < 0.0001 vs. *P*. *longifolia* aqueous extracts; ***: *p* < 0.001 vs. PLLW; **: *p* < 0.01 PLSM vs. PLSW.

**Table 3 molecules-30-04264-t003:** Effects of *Polyalthia longifolia* leaf and stem extracts on HT-29 cell viability.

Plant-Derived Extracts	IC_50_ (95% CI) µg/mL	MEC µg/mL	nH
PLLW	ND	ND	ND
PLLE	33.24 (25.31–44.08)	50	−1.36
PLLM	24.12 (18.32–32.19)	25	−1.11
PLSW	ND	ND	ND
PLSE	56.71 (43.32–76.33)	50	−1.15
PLSM	72.46 (51.66–108.20)	25	−0.76

IC_50_: half maximal inhibitory concentration with 95% confidence intervals (CI); MEC: minimum effective concentration, the lowest concentration showing significant inhibition (*p* < 0.05); nH: Hill slope coefficient. PLLW: *P. longifolia* leaf water extract; PLLE: *P. longifolia* leaf ethanol extract; PLLM: *P. longifolia* leaf methanol extract; PLSW: *P. longifolia* stem water extract; PLSE: *P. longifolia* stem ethanol extract; PLSM: *P. longifolia* stem methanol extract; ND: not determined. All parameters were obtained using the data from Figure 7.

**Table 4 molecules-30-04264-t004:** Identification of compounds in *Polyalthia longifolia* leaf and stem extracts detected by HPLC-DAD analysis.

Compound	Class	Rt (min)	UV-Vis λmax (nm)	PLLW	PLLE	PLLM	PLSW	PLSE	PLSM
Gallic ac.	Phenolic ac.	2.537	270	--	--	+	--	--	--
Chlorogenic ac.	Phenolic ac.	5.509	322	--	--	--	--	--	--
(+)-Catechin	Flavan-3-ol	6.573	280	--	+	+	--	+	+
Epicatechin	Flavan-3-ol	7.019	278	--	+	+	--	+	+
Caffeic ac.	Hydroxycinnamic ac.	7.796	325	+	+	--	+	--	--
Ellagic ac.	Phenolic ac.	7.891	256	+	+++	++	--	+	+
Rosmarinic ac.	Phenolic ac.	9.521	330	--	++	--	--	++	--
Luteolin	Flavonoid	9.813	348	--	+	+	--	--	--
Quercetin	Flavonoid	9.981	257; 376	--	--	--	--	--	--
Kaempferol	Flavonoid	10.753	265; 364	--	--	--	--	+	+
Baicalein	Flavonoid	10.918	277	+	+	+	+	+	+

Compounds were identified by comparing their retention times (Rt) and UV-Vis spectral profiles with those of authentic standards run under identical conditions. The presence and relative concentration of compounds are indicated as: +++: very high concentration; ++: high concentration; +: present; --: not detected; ac: acid. PLLW: *P. longifolia* leaf water extract; PLLE: *P. longifolia* leaf ethanol extract; PLLM: *P. longifolia* leaf methanol extract; PLSW: *P. longifolia* stem water extract; PLSE: *P. longifolia* stem ethanol extract; PLSM: *P. longifolia* stem methanol extract.

## Data Availability

Other experimental data are available in the Appendix A. Additional data are available upon request to authors.

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
