# Peer review of "Exploring the Antidiabetic Properties of Polyalthia longifolia Leaf and Stem Extracts: In Vitro α-Glucosidase and Glycation Inhibition"

_molecules, 2025, doi:10.3390/molecules30214264_

Round 1

Reviewer 1 Report

Comments and Suggestions for Authors

1 Please further confirm the main components in six P. longifolia extracts using techniques such as high-resolution mass spectrometry and provide relevant data results. Meanwhile, analyze the common components and differential constituents to identify the chemical basis for their pharmacological effects.

2 Please conduct content determination of the active constituents to elucidate the dose-effect relationship across the six active fractions.

Author Response

Response to Reviewer 1

1. Summary

Thank you for the time you spent reviewing our manuscript. Your feedback has provided valuable suggestions that help us enhance the quality and clarity of the present research and, furthermore, plan future investigations. We have carefully considered each of your points and have made corresponding revisions to the manuscript.

2. Point-by-point response to Comments and Suggestions for Authors

Comments 1: Please further confirm the main components in six P. longifolia extracts using techniques such as high-resolution mass spectrometry and provide relevant data results. Meanwhile, analyze the common components and differential constituents to identify the chemical basis for their pharmacological effects.

Response 1: We appreciate your suggestion regarding the further chemical characterization of the P. longifolia extracts. We acknowledge that high-resolution mass spectrometry would indeed provide more detailed insights into the extract compositions. However, due to current resource and time constraints, we were unable to perform these additional analyses for this study.

To address your concern, we have conducted a more in-depth phytochemical investigation using the existing HPLC-DAD chromatograms of all six P. longifolia extracts and the eleven standards used in our investigation. We have updated the Methods, Results, and Discussion sections of the manuscript accordingly (lines 356-367, 520-535, 686-689).

Specifically, we have:

-          Provided a more detailed analysis of the common components across the extracts.

-          Highlighted the differential constituents among the extracts.

-          Expanded our discussion on the potential relationships between the identified compounds and the observed pharmacological effects, based on available literature data.

While this approach doesn't provide the level of detail that high-resolution mass spectrometry would, it does offer a more comprehensive view of the extracts' compositions within the scope of our current capabilities. We believe these additions significantly enhance the chemical basis for understanding the pharmacological effects observed in this study.

We have also noted in our experimental plans that a more detailed chemical analysis using high-resolution mass spectrometry would be a valuable next step in this line of research.

We hope these revisions address your concerns and provide a clearer detection of the chemical composition of P. longifolia extracts. Thank you again for your insightful comment, which has helped improve the quality of the manuscript.

Comments 2: Concerning the content determination of active constituents and dose-effect relationships.

Response 2: We appreciate this valuable suggestion. The current study focused on the overall extract activities of P. longifolia leaves and stems rather than their isolated compounds. However, we recognize the importance of understanding the concentration-effect relationships of specific active constituents. To address this point, we have added a paragraph in the Conclusion section (lines 529-535) outlining how future research should study the role of single isolated constituents of the extracts. We suggest that subsequent studies isolate and quantify the major active compounds and evaluate their individual and synergistic effects across different concentrations in the various assays.

Regarding the English language and presentation of our results, we have carefully revised the manuscript to improve clarity and expression throughout. We have paid particular attention to the Introduction, Methods, and Conclusions sections, as per your suggestions. The figures and tables have also been redesigned for better clarity and presentation. Specifically, Table 4 has been repositioned in the text.

We thank you again for your comments, which have provided new ideas for further investigations of P. longifolia. We believe the revisions and clarifications made in our manuscript address your concerns and strengthen the overa

Reviewer 2 Report

Comments and Suggestions for Authors

The manuscript presents an investigation of the alpha-glucosidase inhibitory, antiglycation, and antioxidant activity of three types of extracts (water, ethanol, methanol) obtained from the leaves and stems of Polyalthia longifolia, correlated with their total polyphenol and total flavonoid content. Furthermore, the cytotoxicity of the extracts was evaluated using a human intestinal cell line.

The subject is worthy of investigations, the experiments were carefully designed and performed, the manuscript is well written, and the conclusion are supported by the results. The superior biological activity of the leaf, particularly the methanol extract, was clearly demonstrated.

I have some suggestion and recommendations:

  • Considering that only in vitro tests of alpha-glucosidase inhibition and antiglycation activity were performed, the title should be reconsidered, particularly the terms ‘antidiabetic properties” and “comprehensive in vitro study”.
  • The concentration of acarbose (800 micrograms/ml) was very high compared to samples. The authors should explain why this concentration was chosen and indicate the IC50 value for acarbose under identical experimental conditions.
  • Table 4 contains data only regarding the leaf extracts, while the table caption also mentions the stem extracts; this should be clarified.
  • The authors should perform a quantitative analysis of polyphenols using external calibration. If this is not possible, at least the relative percentage (area percentage) obtained from chromatogram integrations should be provided.
  • The authors should discuss the possibility that compounds other than phenolics contribute to the biological effects of the extracts.
  • Lines 431-432 – the differences observed in TPC and TFC values can also be explained by variations in geographical origin and growing conditions.
  • The authors could better justify the selection of HT-29 cell for cytotoxicity testing and strengthen the discussion on cytotoxic effects in correlation with the other biological activities of the extracts.

Author Response

Response to Reviewer 2

1. Summary

Thank you for your thorough evaluation of the manuscript. The feedback regarding the study design, execution, and overall presentation of this research has been greatly appreciated. Please find the detailed responses below and the corresponding revisions highlighted in the re-submitted files in red characters.  

2. Point-by-point response to Comments and Suggestions for Authors

Comments 1: Considering that only in vitro tests of alpha-glucosidase inhibition and antiglycation activity were performed, the title should be reconsidered, particularly the terms ‘antidiabetic properties” and “comprehensive in vitro study”.

Response 1: We understand your concerns regarding the title of the manuscript, specifically the use of the terms "antidiabetic properties" and "comprehensive in vitro study." Indeed, these phrases may overstate the focus of the research. In light of your feedback, we propose the following revised title:

"Exploring the Antidiabetic Properties of Polyalthia longifolia leaf and stem extracts: in vitro α-glucosidase and glycation inhibition".

This new title more accurately reflects the in vitro tests performed in our research. We believe it addresses your concerns by removing the broader term "comprehensive," and highlighting the exploration of antihyperglycemic-related activity, focusing on the specific targets investigated.

Thus, the title was changed according to your suggestion (page 1, lines 2-4).

Comments 2: The concentration of acarbose (800 micrograms/ml) was very high compared to samples. The authors should explain why this concentration was chosen and indicate the IC50 value for acarbose under identical experimental conditions.

Response 2: We appreciate the reviewer's attention to this important point. The choice of acarbose at 800 μg/mL as a positive control in our investigation was based on several considerations as reported below.

1) Literature precedent: Acarbose is widely recognized as an in vitro and in vivo α-glucosidase inhibitor [1–3].

2) Previous research: In our previous studies, we conducted a detailed potency study of acarbose inhibition, confirming its action as a competitive antagonist of α-glucosidase [3]. The IC50 of acarbose in our α-glucosidase assay protocol (using the enzyme at 0.05 U/mL) was 870.96 µM (562 µg/mL).

3) Ensuring clear inhibition: We chose a concentration above the IC50 (800 μg/mL > 562 µg/mL) to ensure a robust positive control response across all assay replicates.

To improve transparency and reproducibility, we have enhanced the Methods section with a detailed description of the α-glucosidase protocol (page 16, lines 551-561). This addition will allow other investigators to replicate the test under identical conditions.

We hope this explanation clarifies our methodological choices and addresses the reviewer's concerns. Thank you for your thorough evaluation.

Comments 3: Table 4 contains data only regarding the leaf extracts, while the table caption also mentions the stem extracts; this should be clarified.

Response 3: We sincerely thank the reviewer for bringing this inconsistency to our attention. There was indeed a discrepancy between the table caption and the content of Table 4, caused by a formatting error in the manuscript submission process. We have addressed this by including the complete table with both leaf and stem extract data in the revised manuscript. The complete Table 4, now accurately reflecting both leaf and stem data as mentioned in the caption, can be found on page 13 (lines 354-377).

We apologize for any confusion this may have caused and thank you again for bringing this issue to our attention. Your thorough review has helped us improve the accuracy and completeness of our manuscript.

Comments 4: The authors should perform a quantitative analysis of polyphenols using external calibration. If this is not possible, at least the relative percentage (area percentage) obtained from chromatogram integrations should be provided.

Response 4:

We sincerely appreciate the suggestion regarding the quantitative analysis of polyphenols, and understand the importance of providing detailed quantitative information about the polyphenolic content of extracts.

In our current study, Total Phenolic Content (TPC) and Total Flavonoid Content (TFC) assays were provided as a quantitative assessment of these compounds. These methods are widely accepted in the field for initial characterization of plant extracts. Both TPC and TFC were detected and results are presented in Figure 5 and Table 2, pages 8-9.

We acknowledge that these methods have limitations compared to a full quantitative analysis using external calibration or chromatogram integrations. However, they provide valuable insights into the overall phenolic and flavonoid content of P. longifolia extracts, which supports our findings on the biological activities. The suggestion for relative percentage analysis from chromatogram integrations and/or a full quantitative analysis using external calibration would provide more precise data on individual polyphenolic compounds. However, conducting such analyses would require a significant extension of our current research plan, including additional experimental work and resources.

A deeper phytochemical analysis will be performed in future investigations, mainly for PLLE and PLLM extracts, which showed higher activity. However, various constituents were identified in our current analysis (Table 4):

-          Gallic acid was detected in leaf methanol extract (PLLM),

-          (+)-Catechin and epicatechin were found in all alcoholic extracts of both leaves and stems,

-          Caffeic acid was identified in leaf water (PLLW), leaf ethanol (PLLE), and stem water (PLSW) extracts,

-          Ellagic acid was detected across all extracts except the stem water extract (PLSW), with particularly high concentrations in PLLE and PLLM,

-          Rosmarinic acid was prominently found in both leaf and stem ethanol extracts (PLLE and PLSE),

-          Among flavonoids, luteolin was detected in low amounts in ethanol and methanol leaf extracts, while baicalein was consistently present across all extracts.

Furthermore, an indication of the amount of single compound are indicated as: +++: very high concentration; ++: high concentration; +: present; --: not detected. The identified compounds are known in the literature as compounds which may contribute to various biological activities, including α-glucosidase inhibition and antiglycation and antioxidant activities supporting our findings on P. longifolia extracts.

In light of this, we have added a statement in our discussion section (Page 12, lines 355-365) recognizing the need for more detailed quantitative analysis in future research. We have also expanded our discussion of the role of catechins, epicatechins, ellagic acid and luteolin, detected mainly in PLLE and PLLM extracts, to better contextualize our findings with the antihyperglycemic activities proposed for the extracts of P. longifolia.

Comments 5: The authors should discuss the possibility that compounds other than phenolics contribute to the biological effects of the extracts.

Response 5: We sincerely appreciate the reviewer's insightful comment regarding the potential contribution of non-phenolic compounds to the biological effects observed in our study. This is indeed an important consideration in the complex matrix of plant extracts. While our study focused primarily on phenolic compounds due to their well-documented antioxidant and antihyperglycemic properties, we acknowledge that other classes of phytochemicals could also play significant roles in the observed biological activities of P. longifolia extracts. Previous studies on P. longifolia have reported the presence of various non-phenolic compounds, including terpenes (e.g., β-sitosterol and lupeol), alkaloids (e.g., liriodenine), saponins and others, as also mentioned in the introduction on page 2 lines 55-57 [4,5]. These compounds, alone or in synergy with phenolics, could contribute to the overall biological activities observed in our extracts. To address this important point, we have added a discussion in our manuscript (page 12, lines 355-367) acknowledging the potential role of non-phenolic compounds and the need for further comprehensive phytochemical analysis. We have also emphasized that the biological effects observed in our study likely result from the complex interplay of various phytochemicals present in the extracts, not solely from phenolic compounds. In future investigations, we plan to conduct a more comprehensive phytochemical profiling of P. longifolia extracts, particularly focusing on PLLE and PLLM, which showed the highest activities in our current study.

We thank the reviewer for this valuable suggestion, which has helped us provide a more comprehensive discussion of our results and outline important directions for future research.

Comments 6: Lines 431-432 – the differences observed in TPC and TFC values can also be explained by variations in geographical origin and growing conditions.

Response 6:

We sincerely appreciate the reviewer's insightful comment. We agree that geographical origin and growing conditions can significantly influence TPC and TFC values in plant extracts. To address this important point, we have revised our discussion in the manuscript (lines 468-472) to include this consideration.

Comments 7: The authors could better justify the selection of HT-29 cell for cytotoxicity testing and strengthen the discussion on cytotoxic effects in correlation with the other biological activities of the extracts.

Response 7:

Thank you for the valuable suggestion. To address this point, we have expanded the results, discussion, and conclusion sections in the manuscript:

-                     HT-29 cells were chosen due to their relevance in assessing orally administered compounds. As a human colorectal adenocarcinoma cell line, HT-29 cells mimic mature intestinal cells, making them an excellent model for studying potential cytotoxicity on human intestinal epithelium (page 10, lines 291-297).

-                     We noted that the extracts showing the highest α-glucosidase inhibition (PLLE and PLLM) also demonstrated antiproliferative activity on HT-29 cells, albeit at higher concentrations. Overall, the data suggested a potential therapeutic window for α-glucosidase inhibition and cell viability inhibition (lines 566-576, and 669-672).

These additions provide a deeper analysis of the relationship between cytotoxicity and biological activities observed in our study highlighting the potential for developing safe and effective use of P. longifolia extracts, and also to conduct new investigations. Thank you for the valuable suggestions.

References

  1. Pan, G.; Lu, Y.; Wei, Z.; Li, Y.; Li, L.; Pan, X. A Review on the in Vitro and in Vivo Screening of α-Glucosidase Inhibitors. Heliyon 2024, 10, e37467, doi:10.1016/j.heliyon.2024.e37467.
  2. Dirir, A.M.; Daou, M.; Yousef, A.F.; Yousef, L.F. A Review of Alpha-Glucosidase Inhibitors from Plants as Potential Candidates for the Treatment of Type-2 Diabetes. Phytochem Rev 2022, 21, 1049–1079, doi:10.1007/s11101-021-09773-1.
  3. Djeujo, F.M.; Ragazzi, E.; Urettini, M.; Sauro, B.; Cichero, E.; Tonelli, M.; Froldi, G. Magnolol and Luteolin Inhibition of α-Glucosidase Activity: Kinetics and Type of Interaction Detected by In Vitro and In Silico Studies. Pharmaceuticals 2022, 15, 205, doi:10.3390/ph15020205.
  4. Ibrahim, R.B.; Usman, L.A.; Oladiji, E.O.; Adebola, A.O.; Akande, M.O. Gas Chromatography-Mass Spectroscopic Profile, In Vitro Antidiabetic and Radical Scavenging Potentials of Polyalthia Longifolia Leaf Phenolic Extract. NJPAS 2025, 5132–5142, doi:10.48198/NJPAS/22.B07.
  5. Sari, D.P.; Ninomiya, M.; Efdi, M.; Santoni, A.; Ibrahim, S.; Tanaka, K.; Koketsu, M. Clerodane Diterpenes Isolated from Polyalthia Longifolia Induce Apoptosis in Human Leukemia HL-60 Cells. J. Oleo Sci. 2013, 62, 843–848, doi:10.5650/jos.62.843.

Reviewer 3 Report

Comments and Suggestions for Authors

This version has been corrected, attending to the previous comments; however, you need to revise in your manuscript and figures the words for P. longifolia, in vivo and in vitro and change to cursive style

In Figure 7, the reference compound as a positive control is missing. You should include it.

I will suggest to the editor that your manuscript could be published once you make the changes

Author Response

Response to Reviewer 3

1. Summary

Thank you very much for taking the time to review our manuscript. Please find the detailed responses below and the corresponding revisions in red track changes in the re-submitted files.

2. Point-by-point response to Comments and Suggestions for Authors

Comments 1: This version has been corrected, attending to the previous comments; however, you need to revise in your manuscript and figures the words for P. longifolia, in vivo and in vitro and change to cursive style

Response 1: Thank you for your careful review and attention to detail. The feedback on the formatting of specific terms in the manuscript is appreciated. The following changes were implemented as per your suggestion: a) The manuscript was revised to ensure that "P. longifolia" is written in italics throughout the text and figures; b) Regarding the terms "in vitro" and "in vivo", it should be noted that the journal "Molecules" has chosen not to use italics for these terms. Therefore, they remain in regular font according to the style of the journal guidelines. The manuscript, figures, and tables were carefully checked to ensure consistent application of the formatting changes. Thank you again for bringing this to attention. The changes will improve the overall presentation and adherence to both scientific writing conventions and the journal's specific requirements.

Comments 2: In Figure 7, the reference compound as a positive control is missing. You should include it.

Response 2: Thank you for your valuable suggestion regarding Figure 7. We acknowledge the absence of a positive control in the figure. We used luteolin as a positive control at a concentration of 10 µg/mL, which reduced HT-29 viability to 32.54 ± 1.14% (n = 6). This data was not included in the figure because luteolin is an isolated compound, and direct comparison with the extracts could be potentially misleading.

However, following your recommendation, we have added this information to both the results and materials sections of the manuscript. Specifically, the changes can be found on page 10, lines 296-297; and page 20, lines 634-636 in red characters.

The authors are grateful for the thorough review and constructive feedback provided. These revisions have enhanced the accuracy and completeness of the manuscript. We appreciate your valuable suggestions and believe they have contributed to improving the overall quality of our manuscript.

Reviewer 4 Report

Comments and Suggestions for Authors

INTRODUCTION

“…Furthermore, the authors reported that the extracts were able to moderately reduce the activity of α-glucosidase and α-amylase enzymes in vitro [12,13]…”

-Please indicate the plant parts [12,13].

Plant material (leaves and stems): reviewing the literature, it can be seen that several parts of the plant were generally investigated or traditionally used for medicinal purposes.

I suggest you justify the choice of leaves and stems for your research. If the choice is based on preliminary trials (or preliminary screening), please explain.

MATERIALS AND METHODS

Plant material include leaves and stems, or leaves and twigs? Please clarify.

I consider that the extraction yields were low, mainly with aqueous solvent and stem. Normally, 7 days at room temperature are used for a good recovery of phytochemicals from the plant matrix (water). Should you continue working with Polyalthia longifolia extracts, I would suggest optimizing extraction parameters (time, temperature) or exploring alternative extraction methods.

RESULTS AND DISCUSSION

You should compare the results obtained with previous reports in the literature for the species, genus, or family. Or you could compare the results with other species previously investigated as sources of anti-diabetes compounds.

Reviewing your in vitro techniques, it can be seen that specific reaction times were used for each assay. If you make comparisons, remember to do so with extracts tested under similar conditions (reaction times, for example).

Alpha-glucosidase inhibition and ribose-induced BSA glycation…: you could include the IC50 data of the controls.

DISCUSSION

You do not present a discussion on which compounds identified by HPLC could significantly contribute to the action of polar P. longifolia extracts as antidiabetics.

I recommend reviewing the literature in search of other active phytochemicals as alpha-glucosidase inhibitors (potentially present in your extracts). Phenolic compounds are generally bioactive. However, plant extracts may have other groups of compounds that exhibit high anti-diabetes action.

Methanol can be an excellent solvent for recovering phytochemicals. However, its use must consider the risks associated with its toxicity. Please clarify this in your discussion.

Figure 5.C, 5.D:

Please modify the scale (QE mg/g extract)

Author Response

Response to Reviewer 4

1. Summary

Thank you for your detailed review of our manuscript. Your suggestions have been precious in improving our research. We have carefully considered each point and made corresponding revisions to the manuscript. Please find the detailed responses below, with changes highlighted in the re-submitted manuscript.

2. Point-by-point response to Comments and Suggestions for Authors

Comments 1: INTRODUCTION

“…Furthermore, the authors reported that the extracts were able to moderately reduce the activity of α-glucosidase and α-amylase enzymes in vitro [12,13]…”

-Please indicate the plant parts [12,13].

Response 1: Thank you for your suggestion. Providing information about the plant parts used is indeed important, as it relates to the activity reported in the cited studies and enhances the clarity and context of the introduction. To address this point, the related sentence was changed as follows: "Furthermore, the authors reported that the leaf extracts were able to moderately reduce the activity of α-glucosidase and α-amylase enzymes in vitro [12,13]." The modification is on page 2, line 63 of the revised manuscript.

Comments 2: Plant material (leaves and stems): reviewing the literature, it can be seen that several parts of the plant were generally investigated or traditionally used for medicinal purposes.

I suggest you justify the choice of leaves and stems for your research. If the choice is based on preliminary trials (or preliminary screening), please explain.

Response 2: Indeed, various parts of P. longifolia have been investigated or traditionally used for medicinal purposes, such as leaves, stems, roots, seeds, and barks. In the current investigation, the choice to focus on leaves and stems was based on several factors:

-          Leaves and stems are renewable resources that can be harvested without destroying the entire plant, making them more sustainable tissues for potential long-term use.

-          In traditional medicine, leaves of P. longifolia are used to prepare remedies for various ailments, including the treatment of diabetes-related symptoms [1].

-          Previous studies have reported promising antidiabetic activities in leaf extracts of P. longifolia [2,3]. Meanwhile, stems have been less extensively studied, presenting an opportunity for new findings.

-          By studying two types of tissues, leaves and stems, a comparative analysis was conducted, offering insights into the activity and distribution of bioactive compounds within the plant.

The introduction was revised to highlighting the type of tissues chosen based of the literature (page 2, lines 76-80).

We thank the reviewer for highlighting this point, which has allowed us to clarify the methodological choices and enhance the overall context of the current study.

Comments 3: MATERIALS AND METHODS

Plant material include leaves and stems, or leaves and twigs? Please clarify.

Response 3: We sincerely thank the reviewer for bringing this issue to our attention. The present study used both leaves and stems of Polyalthia longifolia. Specifically:

-          Leaves: Mature, fully expanded leaves were collected from the plant.

-          Stems: Young, woody stems (approximately 0.5-1 cm in diameter) were collected. These are sometimes referred to as "twigs" in botanical literature, but the term "stems" is generally used in English articles for consistency in the literature.

To address this point and prevent any misunderstanding, the following change in the revised manuscript was made. In the Materials and Methods section (lines 542-543), a more detailed description of the plant parts was added: "Mature leaves and young woody stems (0.5-1 cm in diameter) of Polyalthia longifolia were collected..."

We thank the reviewer for bringing this point to our attention, as it allows us to provide more precise information about the plant material collected to obtain the extracts, which is crucial for the reproducibility of our study.

Comments 4: I consider that the extraction yields were low, mainly with aqueous solvent and stem. Normally, 7 days at room temperature are used for a good recovery of phytochemicals from the plant matrix (water). Should you continue working with Polyalthia longifolia extracts, I would suggest optimizing extraction parameters (time, temperature) or exploring alternative extraction methods.

Response 4: We sincerely appreciate the suggestion regarding the extraction yields, particularly for the aqueous solvent and stem extracts. Indeed, the yields were lower than expected, especially for the aqueous extractions. The suggestion to optimize extraction parameters and/or explore alternative extraction methods is valuable and well-taken. Certainly, a longer extraction time, such as 7 days at room temperature for aqueous extractions, could potentially improve the recovery of phytochemicals from the P. longifolia tissues. In the current study, the extraction protocol was based on previous work with similar plant materials and was designed to balance extraction efficiency with practical considerations, such as time and resource constraints. However, we recognize that this may not have been optimal for P. longifolia, particularly for aqueous extractions. We thank the reviewer for this valuable suggestion. It provides an important direction for future research with P. longifolia extracts. In future investigations, optimized extraction protocols will be explored to enhance the yield and potentially the bioactivity of the extracts.

Comments 5: RESULTS AND DISCUSSION

You should compare the results obtained with previous reports in the literature for the species, genus, or family. Or you could compare the results with other species previously investigated as sources of anti-diabetes compounds.

Response 5:

The reviewer's suggestions to compare the current results with previous reports in the literature are valuable and provide important context for the findings. While some comparisons were already present in the text (lines 424-436), additional comparisons have been added to enhance the manuscript.

Specifically, the following addition has been made:

“In general, the literature data show that ethanol and methanol extracts from leaves of several plants, including Annonaceae species, are mostly active against α-glucosidase activity, suggesting their potential in diabetes mellitus treatment [47–49]. Our findings with P. longifolia align with this trend, as we observed significant α-glucosidase inhibition with both methanol and ethanol leaf extracts.”

These additions provide a more comprehensive analysis of the results within the context of existing literature on antidiabetic plant compounds.

We thank the reviewer for the valuable suggestion that has helped to deepen the discussion of our results.

Comments 6: Reviewing your in vitro techniques, it can be seen that specific reaction times were used for each assay. If you make comparisons, remember to do so with extracts tested under similar conditions (reaction times, for example).

Response 6:

We sincerely appreciate the reviewer's suggestion drawing attention to the specific experimental protocols, including temperature, time, and other conditions. We used methods adhering to standard protocols with minor modifications necessary to perform the experiments in our laboratory. The necessary references for the methods used are reported for each assay performed in the materials and methods section of the manuscript. We have always tried to compare our data with those reported in the literature under similar conditions, when possible. As mentioned, in the methods section, an appropriate reference is reported for each method. Thank you again for the valuable comment.

Comments 7: Alpha-glucosidase inhibition and ribose-induced BSA glycation…: you could include the IC50 data of the controls.

Response 7:

We appreciate the reviewer's suggestion regarding the inclusion of IC50 data for the controls in the α-glucosidase inhibition and ribose-induced BSA glycation assays. In our experimental design, positive controls were primarily used to verify the performance of each assay, utilizing well-recognized active compounds. For this purpose, in most experimental investigations, it is typical to use a single, effective concentration of the control compound rather than performing a full concentration-effect analysis in each experimental setup. This approach is common in many bioassay protocols and allows us to confirm the validity of the assay while focusing the detailed analysis on the test compounds.

Specifically:

-          For the α-glucosidase inhibition assay, acarbose at 800 μg/mL was used as a positive control, consistently producing significant inhibition.

-          In the ribose-induced BSA glycation assay, aminoguanidine at 2.5 mM served as the positive control, effectively inhibiting glycation.

These concentrations were chosen based on literature values and our previous experience in laboratory setups, ensuring robust inhibition that validates the assay performance. The specific conditions are detailed in each method section, and the effects of the positive controls are reported in Figure 1 for the α-glucosidase inhibition assay, and Figure 4 for the ribose-induced BSA glycation assay.

While we acknowledge that IC50 values for these controls could provide additional context, our focus was on determining the IC50 values of the P. longifolia extracts, which are the novel aspect of our study.

Comments 8: DISCUSSION

You do not present a discussion on which compounds identified by HPLC could significantly contribute to the action of polar P. longifolia extracts as antidiabetics.

Response 8:

We sincerely appreciate the reviewer's insightful comment regarding the discussion of specific compounds identified by HPLC that could significantly contribute to the antidiabetic action of polar P. longifolia extracts. We recognize that this is an important aspect that deserves more attention in our discussion.

To address this point, we have expanded both the result and discussion section (lines 356-368, and 512-535). The HPLC analysis of P. longifolia extracts revealed several phenolic compounds that could contribute to their observed antidiabetic effects. Notably, catechins and epicatechins were found in all alcoholic extracts of both leaves and stems. These compounds have been reported to exhibit α-glucosidase inhibitory activity and antioxidant properties in previous studies (lines 521-524). Ellagic acid, detected in high concentrations in PLLE and PLLM extracts, has also been associated with antidiabetic effects, including α-glucosidase inhibition and reduction of postprandial glucose levels. Furthermore, the presence of gallic acid in the leaf methanol extract (PLLM) is noteworthy, as this compound has demonstrated potent α-glucosidase inhibitory activity in other studies. The flavonoids luteolin and baicalein, detected in our extracts, have also been reported to possess antidiabetic properties, including α-glucosidase inhibition and protection against glycation.

While our study does not establish a direct causal relationship between these compounds and the observed antidiabetic effects, their presence in the most active extracts (PLLE and PLLM) suggests that they may play a significant role in the biological activities we observed. Future studies involving isolation and testing of these individual compounds could provide more definitive evidence of their contribution to the antidiabetic properties of P. longifolia extracts."

We thank the reviewer for highlighting this important aspect, which has allowed us to provide a more comprehensive discussion of the potential active compounds in our extracts and their relevance to the observed antidiabetic effects.

Comments 9: I recommend reviewing the literature in search of other active phytochemicals as alpha-glucosidase inhibitors (potentially present in your extracts). Phenolic compounds are generally bioactive. However, plant extracts may have other groups of compounds that exhibit high anti-diabetes action.

Response 9:

We sincerely appreciate the reviewer's recommendation to consider other active phytochemicals as potential α-glucosidase inhibitors in our extracts. To address this important point, we have expanded our discussion section (lines 530-536).

We thank the reviewer for this valuable suggestion, which has allowed us to provide a more comprehensive discussion of potential active compounds in our extracts and highlight important directions for future research.

Comments 10: Methanol can be an excellent solvent for recovering phytochemicals. However, its use must consider the risks associated with its toxicity. Please clarify this in your discussion.

Response 10:

We sincerely appreciate the reviewer's important observation regarding the use of methanol as an extraction solvent. We agree that while methanol is an excellent solvent for recovering phytochemicals, its use must be carefully considered due to associated toxicity risks.

In light of these considerations, we also tested ethanol and aqueous extracts of P. longifolia, which are more commonly used in traditional medicine and are more compatible with human consumption. Remarkably, the ethanol leaf extract (PLLE) demonstrated very good inhibition of α-glucosidase, inhibition of protein glycation, and antioxidant activity, similar to the methanol extract. These results suggest that ethanol could be a safer alternative for extracting bioactive compounds from P. longifolia for potential applications in phytotherapy.

Future studies should focus on optimizing extraction methods using food-grade solvents to ensure both efficacy and safety in potential antidiabetic preparations from P. longifolia leaves."

We thank the reviewer for this valuable comment, which has allowed us to address an important safety consideration regarding this investigation.

Comments 11: Figure 5.C, 5.D:

Please modify the scale (QE mg/g extract)

Response 11:

We sincerely appreciate the reviewer's suggestion regarding the scale in Figures 5C and 5D. We would like to clarify that the scale for QE mg/g extract (C and D) already has a different range (from 0 to 20 QE) compared to the GAE mg/g extract scale used for total phenolic content (TPC). This difference in scale was intentionally chosen because the values of Total Flavonoid Content (TFC) are smaller than the TPC values.

To provide a comprehensive view of our data:

  • The current scale in Figures 5C and 5D allows for clear visualization of the differences in flavonoid content between extracts.
  • Detailed numerical data for both TPC and TFC are reported in Table 2, allowing for precise comparisons.

We believe this approach balances the need for visual clarity in the figures with the provision of precise numerical data in the table. However, if the reviewer feels that further adjustments would enhance the presentation of the data, we are open to suggestions.

Thank you for your attention to detail, which helps ensure the clear and accurate presentation of our results.

Overall. we sincerely appreciate your valuable feedback, which has significantly improved the quality and depth of our manuscript.
